# Comparing Desktop vs. Mobile Interaction for the Creation of Pervasive Augmented Reality Experiences

**DOI:** 10.3390/jimaging8030079

**Published:** 2022-03-18

**Authors:** Tiago Madeira, Bernardo Marques, Pedro Neves, Paulo Dias, Beatriz Sousa Santos

**Affiliations:** Institute of Electronics and Informatics Engineering of Aveiro (IEETA), Department of Electronics, Telecommunications and Informatics (DETI), University of Aveiro, 3810-193 Aveiro, Portugal; bernardo.marques@ua.pt (B.M.); miguelneves@ua.pt (P.N.); paulo.dias@ua.pt (P.D.); bss@ua.pt (B.S.S.)

**Keywords:** human–computer interaction, interaction paradigms, pervasive augmented reality, 3D reconstruction, user study

## Abstract

This paper presents an evaluation and comparison of interaction methods for the configuration and visualization of pervasive Augmented Reality (AR) experiences using two different platforms: desktop and mobile. AR experiences consist of the enhancement of real-world environments by superimposing additional layers of information, real-time interaction, and accurate 3D registration of virtual and real objects. Pervasive AR extends this concept through experiences that are continuous in space, being aware of and responsive to the user’s context and pose. Currently, the time and technical expertise required to create such applications are the main reasons preventing its widespread use. As such, authoring tools which facilitate the development and configuration of pervasive AR experiences have become progressively more relevant. Their operation often involves the navigation of the real-world scene and the use of the AR equipment itself to add the augmented information within the environment. The proposed experimental tool makes use of 3D scans from physical environments to provide a reconstructed digital replica of such spaces for a desktop-based method, and to enable positional tracking for a mobile-based one. While the desktop platform represents a non-immersive setting, the mobile one provides continuous AR in the physical environment. Both versions can be used to place virtual content and ultimately configure an AR experience. The authoring capabilities of the different platforms were compared by conducting a user study focused on evaluating their usability. Although the AR interface was generally considered more intuitive, the desktop platform shows promise in several aspects, such as remote configuration, lower required effort, and overall better scalability.

## 1. Introduction

Nowadays, additional information can be required to aid with certain tasks, as well as to enhance educational and entertainment experiences. A possible solution to support these activities is the use of Augmented Reality (AR) technologies, allowing for easier access to, and better perception of, additional layers of content superimposed with reality [1,2,3,4]. AR experiences consist of the enhancement of real-world environments through various modalities of information, namely visual, auditory, haptic, and others. In our work, we focus on the overlaying of computer-generated graphics in the scene [5,6,7]. This entails the combination of real and virtual worlds, real-time interaction, and accurate 3D registration of virtual and real objects [8,9,10]. In recent times, AR technology has become more prevalent within numerous domains, such as advertising, medicine, education, robotics, entertainment, tourism, and others. Despite their many advantages, usually these types of solutions are tailored to handle unique and specific use cases [11].

Pervasive AR extends this concept through experiences that are continuous in space, being aware of and responsive to the user’s context and pose [12,13]. It has the potential to help AR evolve from an application with a unique purpose to a multi-purpose continuous experience that changes the way users interact with information and their surroundings.

The creation of these experiences poses several challenges [8,11,12,13,14], namely the time and technical expertise required to create such content, which are the main reasons preventing its widespread use; tracking of the user, i.e., understanding the user pose in relation to the environment; content management, i.e., making different types of digital content available (e.g., text, images, videos, 3D models, sound); adequate configuration of virtual content, i.e., configuring the experiences by choosing where augmented content is placed in the real world, as well as adequate mechanisms to display, filter, and select the relevant information at any given moment; uninterrupted display of information, i.e., making these continuous experiences available in order for them to be explored by the target audiences through adequate mechanisms; suitable interaction, i.e., providing usable interaction techniques to the devices being used, allowing easy-to-use authoring features; accessibility, which must be ensured for inexperienced users, while also providing experts with the full potential of the technology. Lastly, given the effects of the COVID-19 pandemic, which have abruptly upended normal work routines, it seems more relevant than ever to also consider remote features for these emerging experiences. All these key challenges arise, since the creation of a solution capable of streamlining the process of configuring a Pervasive AR experience is still a relatively unexplored topic. Therefore, authoring tools which facilitate the development of these experiences are paramount [15,16,17,18,19,20].

In this paper, we evaluate and compare different interaction methods to create and explore pervasive AR experiences for indoor environments. We take advantage of a 3D reconstruction of the physical space to perform the configuration on a desktop and on a mobile device. Both approaches were evaluated through a user study, based on usability and usefulness. The main goal of this work is to evaluate the feasibility and main benefits of each method for the placement of virtual content in the scene to create an experience that is continuous in space, by tracking and responding to the user’s pose and input. We attempt to analyze and understand the advantages and drawbacks of each one, considering efficiency, versatility, scalability, and usability, and discussing the potential and use cases for each type of solution, facilitating future development and comparisons.

The remainder of this paper is organized as follows: Section 2 introduces some important concepts and related work; Section 3 describes the different methods of the developed prototype; Section 4 presents the characteristics of the user study; Section 5 showcases the obtained results as well as discussion about them. Finally, concluding remarks and future research opportunities are drawn in Section 6.

## 2. Related Work

In order to promote AR growth and its spread to mainstream use, there is a need to increase augmented content and applications. As the quantity and quality of available AR experiences expand, so does their audience, which helps push the boundaries of its development. Authoring tools help the creation and configuration of AR content and are crucial to facilitate the stream of new applications made by a wider user base, as well as to improve the quality of experiences developed by experienced professionals [14,21]. In [15], Nebeling and Speicher divided authoring tools into five classes, according to their levels of fidelity in AR/VR, skill and resources requirements, while also taking into consideration their complexity, interactivity, presence of 2D and 3D content and support of programming scripts. The authors highlight the main problems of the existing authoring tools, which make the creation task difficult for both new and experienced developers. They mention the large number of tools available, paired with their heterogeneity and complexity, which pose challenges even to skilled users when they need to design intricate applications that may require multiple tools working together. To overcome these difficulties, studies and advancements have been made to better understand how to create better authoring tools for all skill levels. In [21] the authors emphasize the importance of authoring tools to boost the mainstream use of AR and analyze 24 tools from publications between 2001 and 2015. These were thoroughly discussed by the authors regarding the authoring paradigms, deployment strategies and dataflow models. A broad conclusion drawn is the importance of the usability and learning curves of authoring tools, a barrier that still exists in the present day.

In our work, we aim to provide an analysis and evaluation of different methods of creating pervasive AR experiences using a simple use case consisting of the placement of virtual content in the scene to create an experience that is continuous in space. To this end, we track the environment and allow the user to manipulate and place virtual objects in the scene. Object positioning is amongst the most fundamental interactions between humans and environments, be it a 2D manipulation interface, a 3D virtual environment, or the physical world [22,23]. The manipulation of a 3D object entails handling of six independent axes, or degrees of freedom, three corresponding to movement, and three of rotation. Existing taxonomies of 3D selection/manipulation techniques are available in [24,25,26,27,28,29,30]. In [25], a fine-grained classification of techniques through their chosen methods of selection, manipulation, and release, was first presented, along with further characterization of these groupings. Subsequent works have improved and expanded this classification.

### 2.1. Considerations on 3D Manipulation

It has been argued in [31,32] that positioning an object should be handled as one single action, as demanding the user to consider partial movements along separate axes seems to make it more cumbersome for them to perform the task. Additionally, it has been shown that, in 3D interaction tasks, movements along the Z axis, or depth axis, took longer and were less accurate when compared with movement along the X and Y axes [33,34].

Collision avoidance has been considered beneficial for 3D manipulation. Fine positioning of objects is greatly aided by the ability to slide objects into place with collision detection/avoidance [35]. One given reason for the effectiveness of collision avoidance is that users often become confused when objects interpenetrate one another, and may experience difficulty in resolving the problem once multiple objects are sharing the same space. Moreover, in the real world, solid objects cannot interpenetrate each other. As such, collision avoidance arises as a reasonable default, similarly to contact assumption [36]. Additionally, most objects can be constrained to always stay in contact with the remainder of the environment, according to [31,32]. This is because objects are bound by gravity in most real scenarios the users experience, and as such, contact is also an acceptable default for most virtual environments. This type of contact-based constraint appears to be beneficial, particularly for novice users [37].

### 2.2. 3D Manipulation in a 2D Interface

Manipulation is potentially a 6DOF task, while a standard computer mouse can only facilitate the simultaneous manipulation of 2DOF. To allow full unconstrained manipulation of the position and rotation, software techniques must be employed to map the 2D input into 3D operations. For instance, 3D widgets—see Figure 1—such as “3D handles” [38], the “skitters and jacks” technique [39], or mode control keys. The utilization of 3D widgets is generally the solution found in modeling and commercial CAD software [40]. Handles split the several DOFs, visibly dividing the manipulation into its partial components. Handles or arrows are usually provided for the three axes of movement, and spheres or circles for the three axes of rotation. When using mode keys, the user is able to modify the 2DOF mouse controls by holding a specific key. The main limitation of this type of manipulation technique is that users must mentally decompose movements into an array of 2DOF operations, which map to individual actions along the axes of the coordinate system. This tends to increase the complexity of the user interface and introduce the problem of mode errors. Even though practice should mitigate these problems, software employing this type of strategy is conducive to a steeper learning curve for the user. A different technique can be to use physical laws, such as gravity and the inability of solid objects to inter-penetrate each other, to help constrain the movement of objects. This can also be accomplished by limiting object movement according to human expectations [37]. One approach was presented in [31,32], based on the observation that, in most real-world scenarios, objects in the scene remain in contact with other objects. To emulate this, the movement algorithm makes use of the surfaces occluded by the moving object to determine its current placement, whilst avoiding collision. The movement of objects can be thought of as sliding over other surfaces in the scene.

## 3. Pervasive AR Prototype

The proposed experimental prototype consists of various components that work towards creating Pervasive AR experiences (Figure 2). The reconstruction of an environment can be accomplished through a plethora of methods that involve different technologies. We used Vuforia Area Targets (VAT) (https://library.vuforia.com/features/environments/area-targets.html. Accessed on 10 January 2022) to obtain a model with environment tracking capabilities. Using a 3D scanner, (e.g., Leica BLK 360), a reconstruction of the environment was obtained. This was used as input to create a VAT, i.e., a data set containing information associated with tracking, 3D geometry, occlusion, and collision, that can be imported into Unity (number 5 in Figure 2). Plus, pre-defined virtual models can be selected and manipulated to provide additional layers of information. This process may use different interaction methods, either using a desktop or mobile setting (Figure 3).

The object placement is performed indirectly, as described in [25], from a list of preloaded prefabs. The 3D manipulation uses collision avoidance and contact-based sliding, ensuring that the object being moved remains in contact with other objects [32]. Depth is handled by sliding the object along the nearest surface to the viewer that its projection falls onto. Figure 4 depicts how the 2D mouse motion is mapped to the 3D object movement. The forward mouse movement, along the Y axis, will move the cube along the Z and Y world axes alternatively, depending on contact detection with the surfaces constraining its movement. When using a 6DOF mobile device, the mapping is somewhat more complex. Both the translation in the Y axis and rotation around the X axis are mapped to the Y and Z world axes translation, as shown in Figure 5.

This technique allows us to reduce 3D positioning to a 2D problem. Objects are moved through their 2D projection and can be directly manipulated. Research on this type of approach suggests that it is amongst the most efficient 3D movement techniques and has a flatter learning curve when compared with other common techniques, such as 3D widgets [32]. Additionally, it appears to be well-suited for usage with both 2DOF and 6DOF input devices, making the comparison between solutions easier, given the similar implementations of movement procedures.

An additional fine-tuning phase could be explored, such as the one described in [25]. This would allow users to rotate freely and offset the objects after their placement. However, it was considered to create considerable complexity and provide little added value, since the sets of objects to be placed are already produced with orientation that matches their intended use and realistic placement in reality. Instead, a rotation around the object’s normal axis was added as part of the movement. This was achieved by using a free DOF in the mobile device’s movement for the AR-based interface (around the Z axis) and the scroll wheel in the computer mouse for the 2D interface.

### 3.1. Mobile Method

Several studies describe experiments using 6DOF input devices for positioning and not orientation [24,25,33,34,41]. Our AR interface is similar for practical reasons. The placement of the objects and its visualization are dependent on the orientation of the device relative to the environment and the user. The position of the placed object is obtained through the mechanism of ray casting, using a conjunction of the device 3 DOF movement (its translation within the environment), and 2 of the 3 DOF in the rotation (Roll and Pitch—Figure 6). This leaves only rotation around the Z axis (Yaw), which was mapped to the rotation of the placed object around its normal axis.

The AR-based interface uses the Vuforia Area Targets (VAT) capabilities, leveraging the use of a digital model of the environment for tracking purposes. The placement of virtual objects is achieved by positioning them relative to the Area Target, taking advantage of its known geometry to place the objects against the environment’s surfaces, as seen in Figure 7.

This is accomplished by virtually casting a ray through the centre of the device’s screen onto the detected planes, allowing physical navigation of the environment and aiming anywhere to define the position of the virtual object. The object is placed upright on the surface and its rotation is automatically calculated using the normal of the surface, which is obtained from the ray-casting process. The rotation around the axis of the normal may be adjusted by by tilting the device, rotating it around the Z axis (Figure 8). Finally, the object is placed by taping the screen.

### 3.2. Desktop Method

Most 3D graphics systems use a mouse-based user interface. However, for 3D interaction, this introduces the problem of translating 2D motions to 3D operations. While several solutions exist, most require users to mentally decompose 2D mouse movements to low-level 3D operations, which is correlated with a high cognitive load [42,43,44,45]. Yet, 2D input devices have been shown to outperform 3D devices, in certain 3D positioning tasks, when mouse movement is mapped to 3D object movement in an intuitive way. Through ray casting, the given 2D XY coordinates of the mouse cursor can be used to project a ray from the user viewpoint, through the 2D point on the display, and into the scene. This technique has been reported to work well when used for object selection/manipulation [25,32,37,46,47,48,49].

The desktop-based interface allows the user to navigate a virtual version of the scene using a 2D interface consisting of the classic computer monitor, keyboard for translation, and mouse for rotation of the point of view. It mimics reality by implementing object collision and gravity for the user avatar, in an attempt to make the navigation experience as similar as possible to a real-life scenario. The placement of the objects is analogous to the AR version. A ray is cast through the mouse pointer onto the mesh of the virtual scene. As the mouse cursor is always at the centre of the screen, moving with the camera, it ties up nicely with the AR version.

The scene model utilized was the same as before. This serves as a bridge that connects the locations of the virtual object in both desktop and mobile applications. The configuration of the experiences contains the object poses with respect to the Area Target, so they appear in the correct location in either visualization platform.

## 4. User Study

A user study was conducted to evaluate the usability of the experimental tools in the two available interaction methods and inform the next steps of our research.

### 4.1. Experimental Design

A within-group experimental design was used. The null hypothesis (H0) considered was that the two experimental conditions are equally usable and acceptable to conduct the selected tasks. The independent variable was the interaction with two levels corresponding to the experimental conditions: C1—desktop and C2—mobile. Performance measures and participants’ opinion were the dependent variables. Participants’ demographic data, as well as previous experience with virtual and augmented reality, were registered as secondary variables.

### 4.2. Tasks

We focused on user navigation and content placement using conditions: C1—desktop and C2—mobile. The goal was to position six virtual objects (e.g., notepad, cardboard box, chair, laptop computer, desktop, and book) in the surrounding space (Figure 9 and Figure 10), always in the same order and with the same target poses. The target pose consisted of a semi-transparent copy of the object model, with a red tint outline, placed in the desired final position and rotation. The participants were instructed to find the current target pose and overlap the object in their control, placing it accordingly (Tasks 30s illustrative video: youtu.be/kSj-HnZvJj0 accessed on 10 January 2022).

### 4.3. Measurements

Task performance was collected: time needed to complete all procedures, logged in seconds by the devices used, and differences in position and rotation in relation to the corresponding target pose. These were calculated and registered for both the mobile and desktop applications during each test. Participants’ opinion was gathered through a post-task questionnaire and interview, including: demographic information, the System Usability Scale questionnaire (SUS), the NASA-TLX, as well as questions concerning user preferences regarding each condition’s usefulness and ease of use.

### 4.4. Procedure

Participants were instructed on the experimental setup, the tasks, and gave their informed consent. They were introduced to the prototype and a time for adaptation was provided. Once a participant was familiarized with and felt comfortable enough to use the prototype, the test would begin. Each of the virtual objects and its respective target pose would be shown at a time and, once its placement was completed, the process would be repeated for a total of six objects. All participants performed the task using both conditions in different orders to minimize learning effects. At the end, participants answered the post-task questionnaire and a small interview was conducted to understand their opinion towards the conditions used. The data collection was conducted under the guidelines of the Declaration of Helsinki. Additionally, all measures were followed to ensure a COVID-19-safe environment during each session of the user study.

### 4.5. Participants

We recruited 17 participants (3 female) with various professions, e.g., Master and Ph.D. students, Researchers, and Faculty members from different fields. Only three participants reported having no prior encounters with AR, while the others had some or vast experience.

## 5. Results and Discussion

On average, tests lasted 17 min, including the aforementioned steps (tasks solely took 8 min on average to complete). Overall, participants were quicker and more precise using the desktop condition. The placement task was 2.77 times faster, 4.12 times more precise in translation and 10.08 times in rotation with the desktop condition (on average). The SUS revealed a score of 92 for the desktop and 76 for the mobile condition, implying excellent and good usability, respectively. Additionally, the NASA-TLX questionnaire indicated that the desktop condition had a lower mental and physical demand, while the mobile had more average demands. In addition, participants considered the desktop platform to be more enjoyable, intuitive, accurate, and preferable for frequent use.

The symmetry of some virtual objects, namely the cardboard box, desktop computer, and book (Figure 9b,e,f and Figure 10b,e,f), caused some users to place them in the correct position but with the inverse orientation. In those cases, results were altered to rectify these shortcomings, adjusting the rotation values with a 180º offset. As can be observed in the comparisons from Figure 11, Figure 12 and Figure 13, the desktop method was deemed quicker and more precise. The results in terms of time and accuracy were always better for this method regarding each virtual object placed. In the box plots of the aforementioned figures, we also found that the performance varies according to the objects, demonstrating the difficulty of their manipulation. Larger models, such as the chair and desktop computer (Figure 9c,e and Figure 10c,e), proved to be harder to place accurately, likely due to requiring an increased distance from the device to the desired location to be captured completely on the screen. As the users were farther from the physical surfaces, their movements lost precision in relation to the consequential transformations on the objects.

Both methods were agreed to have potential for improved performance with training or further experience. For the mobile condition, although some users found the object rotation approach intuitive, most struggled with understanding the necessary movement, suggesting that further research in alternative interaction methods may be required. Yet, this condition will likely provide users with a better perception of the augmented environment, since they will essentially be experiencing it during the creation process. The desktop condition results may have been influenced by the participants’ greater experience with computer interfaces. However, no significant errors in the placement of objects for either method occurred. It can also be argued that both platforms may serve as reliable and satisfactory authoring tools according to the use case. While the setup of the interface for the desktop condition is more elaborate, requiring the use of specific equipment and a relatively complex process, it allows for much more flexibility. This version may be used to configure remote environments, avoiding the need to be present at the desired location. This is especially relevant for dynamic environments with multiple users, vehicles, or robots (e.g., logistics warehouse, assembly lines, etc.). Besides the useful remoteness aspect, it could also prove advantageous for handling the creation, or reconfiguration, of larger or simply more experiences in the same time, with less physical effort. Thus, it is thought to have better scalability. We believe these reasons may justify the additional investment in some pervasive AR experience creation use cases, particularly when the environment tracking mechanisms themselves already involve the creation of a 3D replica of the environment. The proposed conditions also appear to be complementary in some way: create an AR experience through the mobile method, and then fine-tune the experience with the desktop platform. The desktop version may also be extended to immersive experiences in VR, allowing direct manipulation using 6DOF controllers to precisely position virtual objects.

## 6. Conclusions and Future Work

In this work, two different methods for the configuration and visualization of pervasive AR experiences were evaluated and compared, one using a mobile device and one using a desktop platform, leveraging a 3D model for interaction. A user study was conducted to help assess the usability and performance of both. The desktop version excels in precision and practicality, benefiting from a 3D reconstruction of the physical space and a familiar interface, providing accurate control and manipulation of virtual content. The mobile version provides a process closer to the final product, granting better perception of how the augmented content will be experienced. Hence, both versions arise as valid solutions, according to the scenario and requirements being addressed, for creating pervasive AR-based experiences which users can configure, explore, interact with, and learn from.

In order to further pursue the true definition of pervasive AR, more work may be carried out to achieve a context-aware application. A future application could display augmented content dynamically as the user is located inside or outside key areas throughout the space, or even depending on the conditions of external factors, such as the status of devices or machinery in the environment, the triggering of alarms, or the time of the day, for example. Novel features may also be added to the existing prototype, considering newer virtual content such as video, charts, animations, and others. We plan to test the prototype with a dynamic industrial scenario, which presents additional challenges, such as more complex environments and handling the operators’ movement, as well as repositioning of physical objects. Therefore, this study will also continue towards the improvement of the authoring features, which enables more complex and interactive AR experiences for the target users. Additionally, it would be interesting to consider different Head Mounted Display (HMD) versions, taking advantage of different interaction/manipulation methods for virtual content placement, which can be used on site or remotely according to the experience context.

## Figures and Tables

**Figure 1 jimaging-08-00079-f001:**
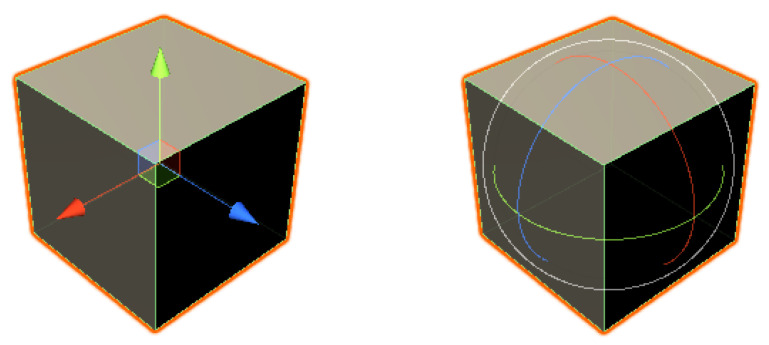
Unity transform gizmos. Position and rotation, respectively.

**Figure 2 jimaging-08-00079-f002:**
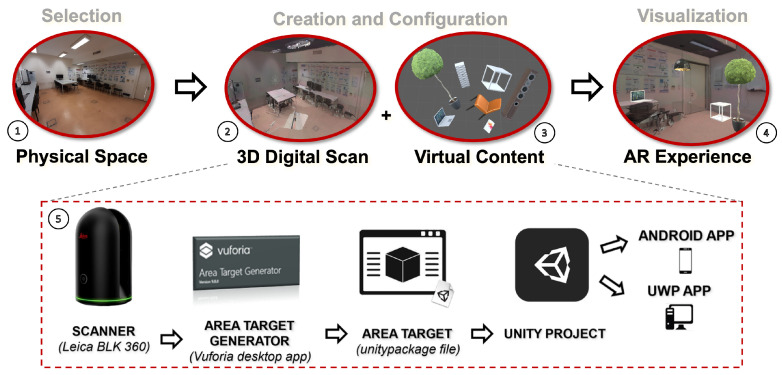
Creation process of pervasive AR experiences: 1. real-world environment; 2. reconstruction of the physical space; 3. integration of pre-existing augmented content according to the experience scope; 4. visualization of continuous AR-based content in the physical environment; 5. overview of the necessary steps to generate the experience.

**Figure 3 jimaging-08-00079-f003:**
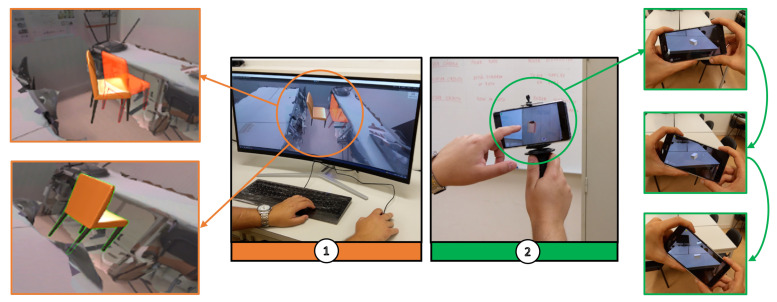
Virtual content configuration: **Left (1)**—VAT model using a desktop computer; **Right (2)**—physical space using a handheld device.

**Figure 4 jimaging-08-00079-f004:**
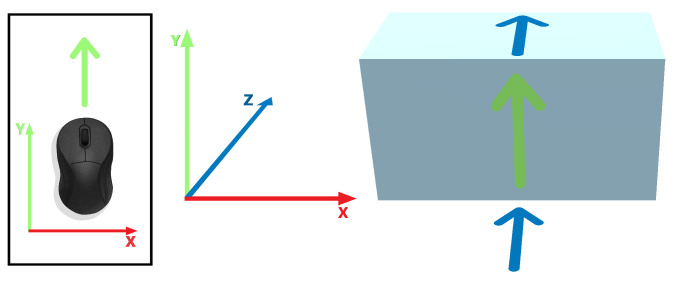
Mouse motion to 3D movement mapping.

**Figure 5 jimaging-08-00079-f005:**
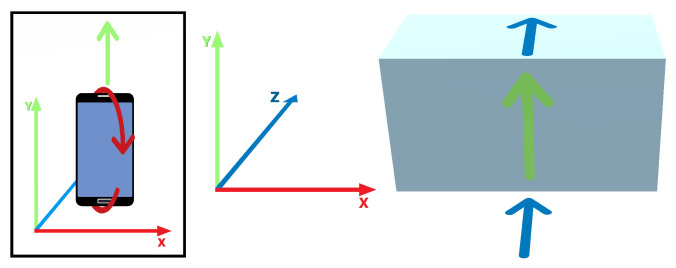
Mobile device motion constrained to 2DOF operation.

**Figure 6 jimaging-08-00079-f006:**
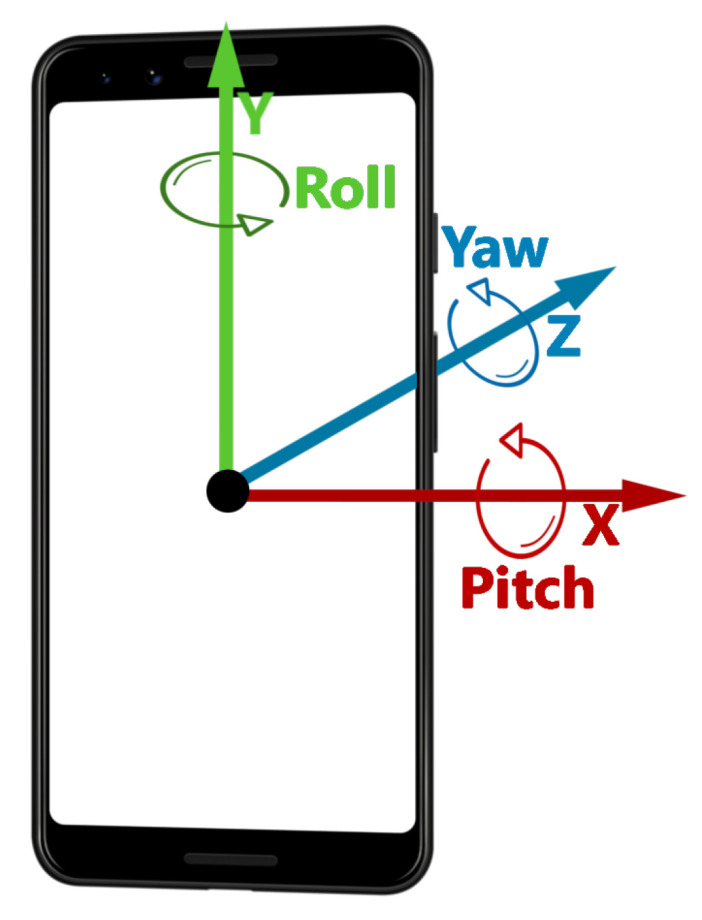
3DOF of Rotation in a smartphone.

**Figure 7 jimaging-08-00079-f007:**
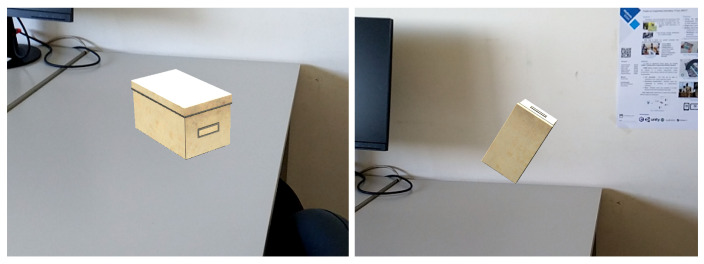
Placement of a virtual box onto a surface according to its normal vector. Depending on where the centre of the screen is aimed, the box base is placed parallel to the table (**left**) or to the wall (**right**) and in contact with the surfaces.

**Figure 8 jimaging-08-00079-f008:**
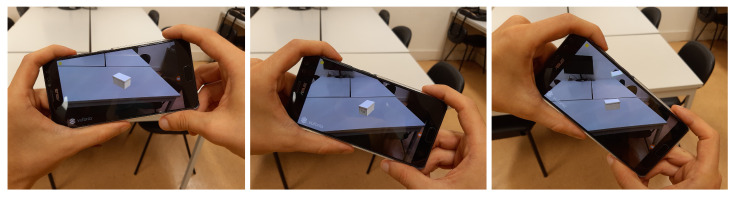
Handling the mobile device to control the virtual object rotation. By turning the device without changing the camera point of view, the object orientation changes while maintaining its position.

**Figure 9 jimaging-08-00079-f009:**
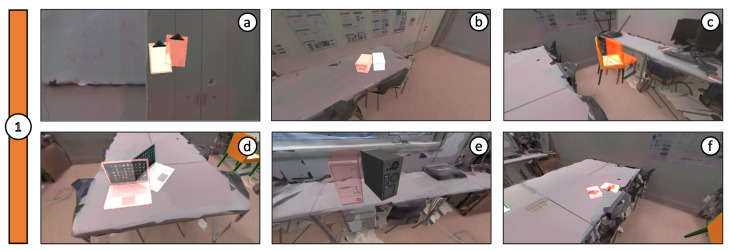
Positioning of 6 virtual objects and their respective target poses (transparent counterpart) using the desktop version: (**a**) notepad; (**b**) cardboard box; (**c**) chair; (**d**) laptop computer; (**e**) desktop; (**f**) book.

**Figure 10 jimaging-08-00079-f010:**
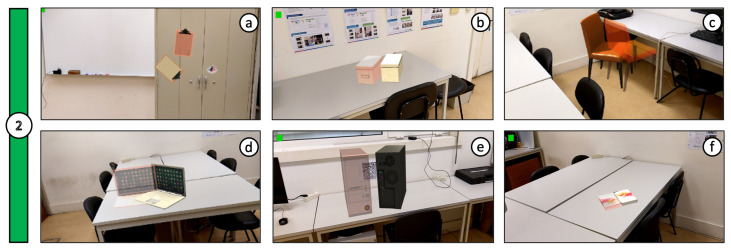
Positioning of 6 virtual objects and their respective target poses (transparent counterpart) through the mobile version: (**a**) notepad; (**b**) cardboard box; (**c**) chair; (**d**) laptop computer; (**e**) desktop; (**f**) book.

**Figure 11 jimaging-08-00079-f011:**
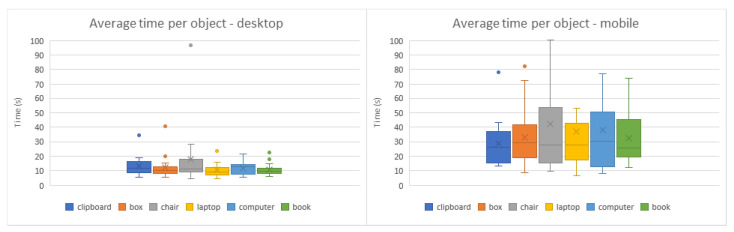
Box plots of the average time it took to place each virtual object in the desktop (**left**) and mobile (**right**) methods.

**Figure 12 jimaging-08-00079-f012:**
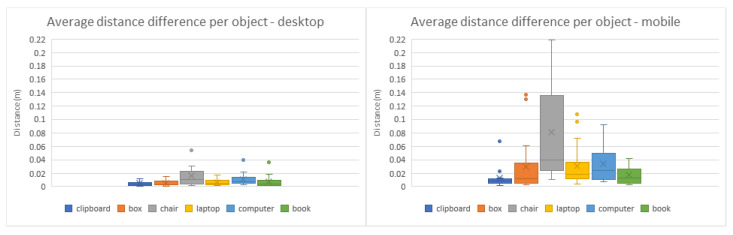
Box plots of the average distance from each virtual object placed to its target position in the desktop (**left**) and mobile (**right**) methods.

**Figure 13 jimaging-08-00079-f013:**
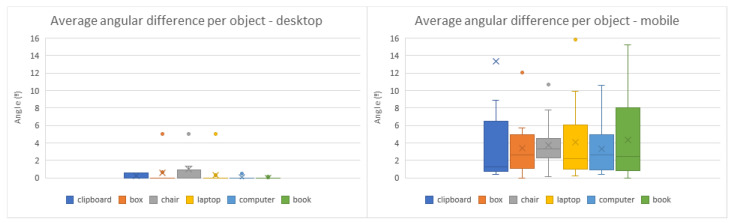
Box plots of the average angular difference from each virtual object placed to its target position in the desktop (**left**) and mobile (**right**) methods.

## Data Availability

Source data are available from the corresponding author upon reasonable request.

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
