# Peer review of "Comparing Desktop vs. Mobile Interaction for the Creation of Pervasive Augmented Reality Experiences"

_2313-433X, 2022, doi:10.3390/jimaging8030079_

Round 1

Reviewer 1 Report

This papers presents the results of a usability study comparing two different approaches to content authoring to be used in augmented reality. Both methods assume that the scene where the augmentation would take place is partialy modeled through scanning and computer vision techniques. Using the model, first method allows the user to place virtual objects using a 2D interface while the second method lets the user do the placement through an AR interface. 

Content authoring is an important problem in AR and these type of methods have been frequently suggested and applied. However, these two of course are not the only possibilities for authoring tasks. These assumes that virtual content is fixed and only their placement in the scene is authored. Building and modifying virtual content especially adding animation (as simple as movement from a point to another) are frequently needed. At least animations should be considered as part of the authoring.

Even with the limited authoring features, the usability setup and experiments is well designed and the results could be useful for some researchers and practioners in the field of AR. Adding animation features to the authoring would increase the interest in the community.

The paper is well written and reading it was enjoyable. The relevant work section is a bit too limited and detailed concentrating only on the placement aspect of the task. Similarly, the technical details about the implementation of desktop and mobile systems is repetitive as these techniques are very well known by the AR practioners. Related sections (most of section 3) can be shortened considerably.

Finaly, the conclusive statement starting on line 314 is a bit problematic. As mentioned above, the study only addresses very limited form of authoring, yet this sentence is generalizing the result to a full-fledged authoring tool. Authors may want to consider revising this sentence.  

Author Response

“This papers presents the results of a usability study comparing two different approaches to content authoring to be used in augmented reality. Both methods assume that the scene where the augmentation would take place is partialy modeled through scanning and computer vision techniques. Using the model, first method allows the user to place virtual objects using a 2D interface while the second method lets the user do the placement through an AR interface.”

We appreciate the reviewer's analysis and constructive comments towards improving the quality of our manuscript. In that sense, some changes were made to the document (marked with the orange color in the new manuscript).

“Content authoring is an important problem in AR and these type of methods have been frequently suggested and applied. However, these two of course are not the only possibilities for authoring tasks. These assumes that virtual content is fixed and only their placement in the scene is authored. Building and modifying virtual content especially adding animation (as simple as movement from a point to another) are frequently needed. At least animations should be considered as part of the authoring. Even with the limited authoring features, the usability setup and experiments is well designed and the results could be useful for some researchers and practioners in the field of AR. Adding animation features to the authoring would increase the interest in the community.”

We thank the reviewer for the suggestion and agree that this would indeed be of interest for our work. The manuscript we present includes only the interactions we considered as the necessary foundation for creating pervasive experiences. Our objective was to create and evaluate a first proposal to place and configure virtual content to gather some insights on useful interaction methods for such goals. In the future, we will certainly expand functionality to support other types of content and interaction as suggested, namely animation authoring, manipulation of additional data, interactive widgets, different views, among others.

“The paper is well written and reading it was enjoyable. The relevant work section is a bit too limited and detailed concentrating only on the placement aspect of the task. Similarly, the technical details about the implementation of desktop and mobile systems is repetitive as these techniques are very well known by the AR practioners. Related sections (most of section 3) can be shortened considerably.”

As per the reviewer’s suggestion, we have made an effort to reduce section 3 by removing some repetition of ideas, as we agree with the reviewer that these techniques are well-known by AR practitioners. Still, we have kept some definitions/information because we believe this will help our work be more accessible and easily intelligible for anyone interested in the field. Additionally, we have added some information about authoring tools and some context about how it relates to our research in the “Related work” section of our manuscript.

“Finally, the conclusive statement starting on line 314 is a bit problematic. As mentioned above, the study only addresses very limited form of authoring, yet this sentence is generalizing the result to a full-fledged authoring tool. Authors may want to consider revising this sentence."

We thank the reviewer for paying close attention. We agree that this part of the text may have been misleading. We have revised it and replaced the text with the following:

We believe these reasons may justify the additional investment in some pervasive AR experience creation use cases, particularly when the environment tracking mechanisms themselves already involve the creation of a 3D replica of the environment. The proposed conditions also appear to be complementary in some way: create an AR experience through the mobile method, and then fine-tune the experience with the desktop platform. The Desktop version may also be extended to immersive experiences in VR, allowing direct manipulation using 6DOF controllers to precisely position virtual objects.

Reviewer 2 Report

The proposed paper presents a prototype intended to compared configuration and visualization aspects of pervasive (context-aware) AR experiences using two different platforms: desktop and mobile. The paper is organized and well documented with recent published related work. The introduction provides sufficient background related to AR experiences used in various domains such as advertising, education, medicine, robotics, entertainment, tourism, etc.  The concept of pervasive AR that enables continuous in space experiences and is aware and responsive to the user`s context and most important pose. The pervasive aspects have the potential to extend the development and use of augmented reality techniques with various domains.

The research presented in the proposed paper is significant as it describes a workflow used to create engaging pervasive augmented reality experiences intended for indoor environments. The authors made use of the well-known augmented reality software developed by Vuforia implemented in Unity to facilitate the development of the Android App and UWP App. For the reconstruction of the environment the authors have used the Vuforia Area Targets (VAT) software solution. To define accurate input data for the Vuforia Area Targets, the authors have used the Leica BLK360 3D scanner to acquire the 3D model of the environment. An accurate 3D model enables accurate 3D tracking, especially when dealing with occlusion and collision aspects associated with pervasive (context-aware) augmented reality experiences. The workflow presented by the authors is illustrated in Figure 2.

The aspects related to the virtual content configuration for both desktop computer and handheld device make use of an object selection that enables the use of prefabs preloaded into the application.

The 3D manipulation technique intended for the desktop computer application makes use of the mouse motion paired with contact-based sliding and collision avoidance. The desktop application uses the classic keyboard and mouse navigation, and it allows the user to navigate the virtual scene defined to mimic the real environment with the help of object collision and gravity elements.

For the mobile handheld device, the authors have used a system that allows the objects to be placed upright using the normal of the user chosen surface obtained using ray-casting technique. The proposed implementation allows the user to adjust the rotation around the axis of the normal tilting the device without changing the camera point of view.

Both applications used the same scene model defined as an Area Target, this 3D reconstructed model is used as a reference to connect the locations of the virtual objects in both desktop and mobile application. The pervasive aspects of the proposed method made use of the Area Target to enable the correct placement and visualization of the 3D prefabs models within the proposed augmented reality applications.

The proposed research methodology and user study conducted by the authors are well documented. The user study involved a total of 17 participants that included both students, researchers, and faculty members. Out of the total number, only 3 participants had no prior experience related with AR.

The presented results highlight the fact that participants were much quicker and more important, more precise using the desktop setup. The authors have defined a SUS (System Usability System) to evaluate the overall usability of the proposed systems, both have obtained a good ratting.

The 3D models used within the application are various indoor objects such as chairs, laptops, computer case, books, boxes and a clipboard. All these objects have a symmetric design, but their overall shapes and orientation can be easily identified. The cardboard box has a cutout section on the width of the model to ensure the correct orientation (Figure 10 b). As presented by the cardboard and other objects caused some users to place them with the inverse orientation.

The presented conclusions are supported by the results. As presented by the authors the mobile application provides better perception on how the augmented content can be experienced. The main advantage of the desktop application is related with both precision and ease of use having the main benefit of a 3D reconstruction of the physical space as well as a familiar navigation and interface.

The authors intend to test the proposed prototype on a more dynamic industrial scenario with both operators’ movement and the repositioning of physical objects. This would be interesting to a wide variety of readers and researchers with the rapid development of technology and smart automation of various industries sectors.

The authors intend to use of various Head Mounted Display to enable immersive experiences in VR and enable the direct manipulation using 6DOF controllers to position virtual objects, there are multiple related work published in this direction and multiple commercially available controllers.

I consider that the use of modern AR eye-wear equipment (such as HoloLens 2 or Moverio BT-40) would be a more suitable future work research direction. This would allow users to intuitive rotate the 3D model around the desired axis on the surface normal using natural hand gestures, therefore eliminating some of the drawbacks of the proposed mobile device AR application.

Author Response

“The proposed paper presents a prototype intended to compared configuration and visualization aspects of pervasive (context-aware) AR experiences using two different platforms: desktop and mobile. The paper is organized and well documented with recent published related work. The introduction provides sufficient background related to AR experiences used in various domains such as advertising, education, medicine, robotics, entertainment, tourism, etc. (...) The presented conclusions are supported by the results. As presented by the authors the mobile application provides better perception on how the augmented content can be experienced. The main advantage of the desktop application is related with both precision and ease of use having the main benefit of a 3D reconstruction of the physical space as well as a familiar navigation and interface.”

We must thank the reviewer for their detailed analysis and positive feedback. We are much obliged for the constructive appreciation of our manuscript.

“The authors intend to test the proposed prototype on a more dynamic industrial scenario with both operators’ movement and the repositioning of physical objects. This would be interesting to a wide variety of readers and researchers with the rapid development of technology and smart automation of various industries sectors. The authors intend to use of various Head Mounted Display to enable immersive experiences in VR and enable the direct manipulation using 6DOF controllers to position virtual objects, there are multiple related work published in this direction and multiple commercially available controllers. I consider that the use of modern AR eye-wear equipment (such as HoloLens 2 or Moverio BT-40) would be a more suitable future work research direction. This would allow users to intuitive rotate the 3D model around the desired axis on the surface normal using natural hand gestures, therefore eliminating some of the drawbacks of the proposed mobile device AR application.”

We appreciate the reviewer’s relevant suggestions. Indeed, these are quite practical ideas that perfectly line up with our planned work, and that we will be trying to implement in the near future.

Reviewer 3 Report

In this paper, two types of prototypes for the configuration and visualization of pervasive AR experiences are proposed, and the usability is evaluated by a user study.

There are several points to be clarified:

1. In the text, the use of this term, "Pervasive" AR experiences, is mixed with the conventional AR experiences, which are not the same.

2. It seems that Related Work jumps into quite narrow scopes without background and relevant works about authoring tools.

3. This study proposes prototypes, but they do not include any novel approaches. Also, it is not clear if the proposed prototypes have distinctive features for "Pervasive" AR experiences.

4. The experiments are quite simple, so it seems that the results are quickly concluded without providing more valuable findings in terms of "Pervasive" AR experiences.

Please clarify these points.

Author Response

“There are several points to be clarified:”

We appreciate the reviewer's comment towards improving the quality of this research. In that sense, changes were made to the document (highlighted by the orange color).

“1. In the text, the use of this term, "Pervasive" AR experiences, is mixed with the conventional AR experiences, which are not the same.”

Thank you for calling our attention to this matter. We have rectified this in the manuscript and now only refer to conventional AR experiences when explaining how they are different from Pervasive AR experiences - in the introduction, and now also in the abstract, in an attempt to make the text more consistent and clear for possible readers.

“2. It seems that Related Work jumps into quite narrow scopes without background and relevant works about authoring tools.”

As per the reviewer’s request, we have added information about authoring tools and some context about how it relates to our research in the “Related work” section of our manuscript (marked in the orange color).

“3. This study proposes prototypes, but they do not include any novel approaches. Also, it is not clear if the proposed prototypes have distinctive features for "Pervasive" AR experiences.”

We realize the importance of the reviewer's comment and took extra care to make the main objective of our work clearer in the manuscript, as reported below:

The main goal of this work is to evaluate the feasibility and main benefits of each method for the placement of virtual content in the scene to create an experience that is continuous in space, by tracking and responding to the user's pose and input. We attempt to analyse and understand the advantages and drawbacks of each one, considering efficiency, versatility, scalability, and usability, and discussing the potential and use cases for each type of solution, facilitating future development and comparisons.

“4. The experiments are quite simple, so it seems that the results are quickly concluded without providing more valuable findings in terms of "Pervasive" AR experiences.”

With this exploratory work, we intended to evaluate the basic authoring features to achieve pervasive AR, allowing for the creation of an experience that is continuous in space, by tracking and responding to the user's pose and input. This can be extended to more complex scenarios, and we plan to include additional functionalities in the future to allow for the use of other types of content and interaction. 

We agree with the reviewer that the experiments considered are simple. However, we believe only after this first stepping stone, i.e., creating a first approach to place and configure content, as well as gathering some insight on useful interaction methods, can we start to advance to more complex use-cases that benefit from the insights gathered and that consider the enrichment of the created experiences. 

We also believe it is important to note that we could not find specific guidelines on how to conduct user studies that take into account the characteristics of  Pervasive AR experiences. Therefore, we have merely used our own experience within this topic. Still, we have tried to conduct, to the best of our abilities, an adequate study in terms of evaluation, although we recognize this may be an interesting topic of further research by the community in the near future.

Round 2

Reviewer 1 Report

The changes made to the first version does not address my concerns fully. I pointed out some shortcomings when addressed can improve the manuscript's impact. Other than revising the writeup of one of the sections, no new experiments or features are added to the study. 

It is just the repeat of my original comments/concerns. In particular:

"... Building and modifying virtual content especially adding animation (as simple as movement from a point to another) are frequently needed. At least animations should be considered as part of the authoring."

The authors have provided nothing in this regard.

Author Response

“The changes made to the first version does not address my concerns fully. I pointed out some shortcomings when addressed can improve the manuscript's impact. Other than revising the writeup of one of the sections, no new experiments or features are added to the study. 
It is just the repeat of my original comments/concerns. In particular: ‘... Building and modifying virtual content especially adding animation (as simple as movement from a point to another) are frequently needed. At least animations should be considered as part of the authoring.’ The authors have provided nothing in this regard.”

We thank the reviewer for the comments and suggestions.

It is important to underline that the main goal of our work is to compare two different interaction methods for configuration and visualization of virtual content on top of real-world environments. The focus of the work is not on the developed prototypes (we do not consider the supported functionality as the novelty of our study at this stage). Our focus is rather on interaction methodologies, as we have now tried to make clearer in the manuscript. For the particular set of tests considered at the moment, animations should follow the same principles as the static virtual content being considered.

While we agree with the reviewer that pervasive AR experiences require additional features such as animations for capturing user’s interest and attention, our main objective with this study is in the comparison of methods being made, in light of the current state of the art. The addition of new features to the proposed prototype would require, not only the software development, but also the preparation and conduction of a new user study, as well as a thorough update of the manuscript. All of this within a very narrow time frame, while adding little value to what we consider to be the main contribution of our work. 

We argue that, at the present time, this manuscript presents a relevant contribution to the state of the art, even without considering animations in the performed user study. Still, we recognize that, in the future, including such type of content in our prototypes and conducting user studies focusing on how users react to the content being presented, during pervasive AR experiences, may be relevant. However, we believe this falls outside the scope of this manuscript.

We have added additional content based on the reviewer's suggestions to our ‘Conclusions and Future Work’ section of the manuscript (highlighted by the orange color).

Reviewer 3 Report

The response is acceptable, but there is one thing that I would like to mention. 

In this revision, the main goal of this paper is clarified, but there are several sentences, still making it slightly vague: 
- In the Abstract, starting with "This paper presents a prototype~" 
- In the Introduction, starting with "In this paper, we propose a solution~" 
- In the Conclusion, starting with "In this work, a prototype~ was proposed."

In your right intention, the prototypes should be presented as experimental tools for evaluations, but in this paper, it sounds like, "they are proposed for pervasive AR experience, and their usability is evaluated". That's why I pointed out it in the previous review; thus, please clarify it overall again.

Author Response

The response is acceptable, but there is one thing that I would like to mention. In this revision, the main goal of this paper is clarified, but there are several sentences, still making it slightly vague: 
- In the Abstract, starting with "This paper presents a prototype~" 
- In the Introduction, starting with "In this paper, we propose a solution~" 
- In the Conclusion, starting with "In this work, a prototype~ was proposed."
In your right intention, the prototypes should be presented as experimental tools for evaluations, but in this paper, it sounds like, "they are proposed for pervasive AR experience, and their usability is evaluated". That's why I pointed out it in the previous review; thus, please clarify it overall again.”

We thank the reviewer once again for the constructive criticism towards improving the quality of our manuscript. We have made an additional effort to properly clarify our goal. As per the reviewer's request, an overall revision of the manuscript was done, paying particular attention to the sentences pointed out, as we agree that they could have been misleading. The changes are highlighted in orange color in the manuscript.

Round 3

Reviewer 1 Report

Latest revisions makes it clear that no further work on this manuscript is planned by the authors. Although I believe the suggested work would improve the impact of the paper, I am okay with their response.